

# Low winter precipitation, but not warm autumns and springs, threatens mountain butterflies in middle-high mountains

Martin Konvicka[1,2], Tomas Kuras[3], Jana Liparova[2], Vit Slezak[4],
Dita Horázná[2], Jan Klečka[2] and Irena Kleckova[2]

[1] Faculty of Science, University of South Bohemia, Ceske Budejovice, Czech Republic
[2] Institute of Entomology, Czech Academy of Sciences, Biology Centre, Ceske Budejovice, Czech Republic
[3] Faculty of Science, Palacký University Olomouc, Olomouc, Czech Republic
[4] Jeseníky Protected Landscape Area Administration, Jesenik, Czech Republic

## ABSTRACT

Low-elevation mountains represent unique model systems to study species endangered by climate warming, such as subalpine and alpine species of butterflies. We aimed to test the effect of climate variables experienced by *Erebia* butterflies during their development on adult abundances and phenology, targeting the key climate factors determining the population dynamics of mountain insects. We analysed data from a long-term monitoring of adults of two subalpine and alpine butterfly species, *Erebia epiphron* and *E. sudetica* (Nymphalidae: Satyrinae) in the Jeseník Mts and Krkonoše Mts (Czech Republic). Our data revealed consistent patterns in their responses to climatic conditions. Lower precipitation (*i.e.*, less snow cover) experienced by overwintering larvae decreases subsequent adult abundances. Conversely, warmer autumns and warmer and drier springs during the active larval phase increase adult abundances and lead to earlier onset and extended duration of the flight season. The population trends of these mountain butterflies are stable or even increasing. On the background of generally increasing temperatures within the mountain ranges, population stability indicates dynamic equilibrium of positive and detrimental consequences of climate warming among different life history stages. These contradictory effects warn against simplistic predictions of climate change consequences on mountain species based only on predicted increases in average temperature. Microclimate variability may facilitate the survival of mountain insect populations, however the availability of suitable habitats will strongly depend on the management of mountain grasslands.

Corresponding author
Irena Kleckova,
irena.slamova@gmail.com

## INTRODUCTION

One of the conservation concerns with the currently warming climate is the fate of alpine-zone species inhabiting "middle-high mountains", which only slightly overtop the timberline (*Boggs & Murphy, 1997*; *Schmitt et al., 2014*). Whereas space allows the alpine species upslope distribution shifts in high mountain ranges (*Pauli et al., 2012*), species

inhabiting lower mountains may become trapped at their current locations (*Freeman et al., 2018*; *Schmitt, Cizek & Konvicka, 2005*), *e.g.*, by ascending timberline (*Roland & Matter, 2007*). It is desirable to closely monitor such systems, to analyse their responses to changing climate, and to design, if necessary, rescue measures, which may include management of habitats (*Garcia-Gonzalez, 2008*) or translocations (*Schmitt, Cizek & Konvicka, 2005*).

With over 100 species across the Holarctic realm and 40 species in Europe (*Tennet, 2008*), the butterfly genus *Erebia* Dalman, 1816 (Nymphalidae, Satyrinae) represents a clade with many of its species adapted to cold conditions. It originated in Central Asia during the Miocene (*Pena et al., 2015*) and radiated into rising mountains (*Hinojosa et al., 2018*). All constituent species develop on grasses, overwinter as larvae, and either form a single annual brood, or prolong their development across multiple seasons (*Sonderegger, 2005*). They display remarkable diversity in larval cold hardiness (*Vrba et al., 2017a*) and adult thermoregulation (*Kleckova, Konvicka & Klecka, 2014*), resulting in a diversity of micro-habitat use (*Sonderegger, 2005*) and diverse species assemblages in high mountains (*Grill et al., 2020*; *Polic et al., 2016*). Lower-altitude mountains host lower *Erebia* diversity but allow exploring the effects of climatic variability on sub-alpine butterflies on the margins of their climatic tolerances. Earlier studies revealed upslope distribution shifts (*Franco et al., 2006*; *Scalercio et al., 2014*) and advancing adult flight (*Konvicka et al., 2016*) attributable to the currently warming climate.

Climate, in particular temperature, affects insect thermoregulation (*Ellers & Boggs, 2004*; *Turlure et al., 2010*), overwintering mortality (*Vrba et al., 2014*), and adult activity (*Buckley & Kingsolver, 2012*). It influences food intake (*Forrest & Thomson, 2011*), development rate (*Ayers & Scriber, 1994*), and pressure from natural enemies (*Corcos et al., 2018*). All these factors ultimately affect adult abundance (*Roland, Filazzola & Matter, 2021*) and flight period timing (*Gutiérrez & Wilson, 2020*).

Adult abundance affects populations' survival across years (*McLaughlin et al., 2002*), dispersal across habitats (*Hanski, 1999*), and adaptation to changing conditions (*Watt et al., 2003*). The links between phenology and population fitness are more intricate (*Davies, 2019*). For mountain species restricted to narrow time windows with suitable conditions, advanced adult flight period prolongs the time available for mating and oviposition (*Stewart et al., 2020*). The physiological triggers speeding-up development may be intuitive, such as warm springs (*Gutiérrez & Wilson, 2020*; *Stewart et al. 2020*), or less straightforward, and may include warm periods in autumn (*Pak, Biddinger & Bjornstad, 2019*) or deep winter frosts (*Stalhandske, Gotthard & Leimar, 2017*). On the other hand, too warm temperatures during larval periods may impair energy assimilation (*Klockmann, Wallmeyer & Fischer, 2018*) and even increase mortality (*Karl et al., 2011*), and adult mortality may increase due to heat shocks (*Janowitz & Fischer, 2011*).

Here, we analyse the effects of climate variability on adult abundance and phenology of two *Erebia* species inhabiting two middle-high massifs of the Sudeten mountains, Czech Republic, Central Europe: the Hrubý Jeseník Mts and Krkonoše Mts. The treeline within these mountain ranges is situated at higher temperatures than the global treeline isotherm (*Kuzelova & Treml, 2020*) so that the low-elevation alpine habitats would be
overgrown by forest, if following the thermal isoclines only. This can be caused by winds (cf. *Kaspar, Hosek & Treml, 2017*), a time-lag of tree growth response to abrupt climate change, or past pastoralism (*Kuzelova & Treml, 2020*). These mountains have recently experienced rising temperatures (*Migala, Urban & Tomczyński, 2016*), whereas precipitation is highly fluctuating, without a long-term trend (*Twardosz & Cebulska, 2020*). The studied butterflies *Erebia epiphron* (Knoch, 1783) and *E. sudetica* (Staudinger, 1861) are similar morphologically, but use different habitats. The former inhabits wind-exposed summit plateaus, whereas the latter occupies wind-shielded valley slopes (*Kuras et al., 2003*). Both species occur naturally in the Jeseník Mts, whereas the Krkonoše Mts host a non-native *E. epiphron* population, established there since the 1930s (*Schmitt, Cizek & Konvicka, 2005*).

Owing to restriction of the species to the narrow sub-alpine/alpine zone, population trends cannot be masked by immigration from lower elevations (cf. *Gutiérrez & Wilson, 2020*). The conditions experienced by earlier developmental stages determine abundance of subsequent phases (*Radchuk, Turlure & Schtickzelle, 2013*; *Roland & Matter, 2016*). Thus, we tested the effect of climatic conditions experienced by larvae during the autumn, winter and spring on subsequent adult abundances. *Erebia* females are in flight and oviposit during August. Hatched larvae feed on grasses during September and October. Then, the larvae overwinter until May and start to activate again in June. It is difficult to quantify the direct effect of climate on larval survival (cf. *Rytteri, 2021*). In our study, we defined two periods of larval activity (autumn and spring) and the overwintering period. Then, we analysed the effect of climatic conditions during these developmental phases on subsequent adult abundances based on long-term adult monitoring data.

We tested which climate variables experienced by *Erebia* butterflies during their development influence their adult abundances and phenology, targeting the key climate factors determining the population dynamics of mountain specialist insects.

## MATERIALS AND METHODS

### Study area and species

Both the Jeseník Mts (highest summit: Praděd, 1,491 m, N 50°4.96′, E 17°13.85′, hereinafter "J") and Krkonoše Mts (Sněžka, 1,603 m, N 50°44.16′, E 15°44.38′ E, hereinafter "K") were formed by Variscan orogeny and consist of rolling ridges built from metamorphic (J, K) and crystallinite (K) rocks. The timberline, situated at ca 1,300 m, is formed by sparse *Picea abies* growths, replaced upslope by *Pinus mugo* shrubs (the latter non-native in J: *Kasak et al., 2015*). The summit elevations are covered by species-poor grasslands and heaths, dominated by *Nardus stricta, Avenella flexuosa, Vaccinium myrtillus*, and *Calluna vulgaris* ($\approx$10 km$^2$ in J, $\approx$50 km$^2$ in K). Much richer and structurally diverse vegetation is found at valley headwalls near the timberline (*Bures, 2018*, Fig. 1). The (sub)alpine fauna of the mountains is impoverished due to isolation from other massifs overtopping the timberline (aerial distance J–K: 130 km; J–Mala Fatra Mts, Carpathians: 160 km; J–Schneeberg Mt, Eastern Alps: 270 km).

*Erebia sudetica* occurs in several mountain systems outside of the Alps: Mons du Cantal, France; the Carpathians (Rodna Mts, Retezat Mts, Godeanu-Tarcu Mts, Ciuicas Mts,

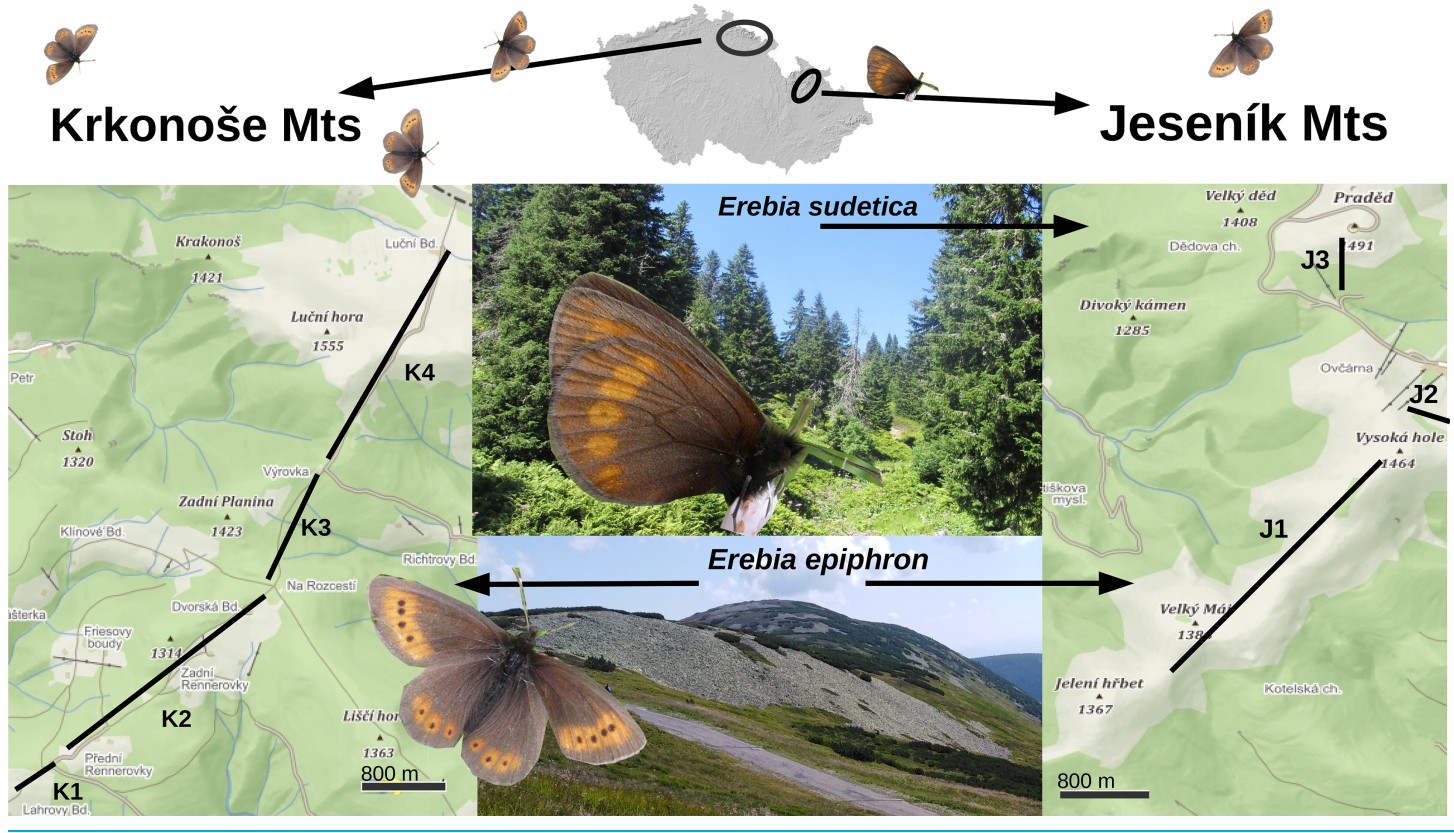

**Figure 1 A map displaying the transects for monitoring *Erebia epiphron* and *E. sudetica* in the Jeseník Mts (J1–J3) and *E. epiphron* in the Krkonoše Mts (K1–K4), Czech Republic.** In the Jeseník Mts, *E. epiphron* occurs at summit grasslands, whereas *E. sudetica* inhabits tall-herb formations around the timberline, situated at ≈1,300 m a.s.l. and formed by dwarf *Picea abies* trees and occasional patches of non-native *Pinus mugo* brushwood. In the Krkonoše Mts, the non-native population of *E. epiphron* descends from the subalpine zone with patches of native *P. mugo* (1,200–1,500 m a.sl.) to cultural meadows (down to 1,100 m a.s.l.). Map source: Mapy.cz (CC-BY-SA 4.0).

Fagaras Mts, *Cuvelier & Dinca, 2007*); Jeseníky Mts, the Czech Republic. It occurs in two Western Alps districts (E. Isére, France; the Bernese Alps, Switzerland). In terms of taxonomy, it appears paraphyletic with *E. melampus* (Fuessly, 1775) (*Haubrich & Schmitt, 2007*; *Pena et al., 2015*) and J represents the northernmost and highly isolated locality of the *melampus/sudetica* species complex. In J, it forms spatially restricted colonies at subalpine tall-herb formations, plus at lower-elevated woodland openings (*Konvicka et al., 2014*).

*Erebia epiphron* is the most widely distributed European alpine *Erebia*. Besides J and K, it inhabits the mountains of Scotland and northern England, Cantabrian Mts in Spain, the Harz Mts in Germany (extinct), the Pyrenees, Massif Central and Vosges in France, Central Apennines in Italy, Alps, Carpathians, and the Dinaric mountains of the Balkans southward to northernmost Greece (*Hinojosa et al., 2018*, see *Minter et al., 2020*). The populations of particular mountain regions are genetically unique (*Minter et al., 2020*). Across this range, it prefers *Nardus* dominated grasslands on nutrient-poor bedrock (*Ewing et al., 2020*). It is restricted to elevations above the timberline in J (*Kuras et al.,*

*2003*), whereas its non-native K population has also colonised cultural grasslands down to ≈1,100 m (*Cizek et al., 2015*).

Both species are in flight in July–August. During warm years, the adults emerge at the end of June. Larvae are oligophagous on grasses (*Sonderegger, 2005*). They feed during September and October, overwinter in grass tussock, and resume feeding after the snow melt in mid-May. Biennial development occurs in *E. epiphron* in the Alps (*Sonderegger, 2005*), whereas univoltine development was observed in *E. melampus* (*Wipking & Mengelkoch, 1994*), closely related to *E. sudetica*. In our study system, we assume that univoltine development prevails also in *E. epiphron*, as the species does not display biennial adult abundance fluctuations in J (*Konvicka et al., 2016*) and both species developed with a single overwintering in a rearing experiment (*Kuras et al., 2001*).

## Adult numbers monitoring

The study is based on transect monitoring of relative abundance of adult butterflies (2009–2020, plus an earlier 5-year period 1995–1999 for one of the transects), combined with meteorological data from nearby weather stations. This time span is sufficient to detect butterfly population trends and to test the effects of climate variability on adult abundance changes (see *Wilson & Fox, 2020*). We established three transects in J (J1–J3, total length 4.2 km, altitude 1,325–1,492 m a.s.l.) and four in K (K1–K4, total length 7.1 km, altitude 1,120–1,510 m a.s.l.) (Fig. 1 and Supporting Information, Supplementary methods).

We monitored both species in J in 2009–2020 (12 years), plus *E. epiphron*, J1 only, in 1995–1999 (5 years). In K, *E. epiphron* was monitored in 2010–2020 (11 years) (Table S1). The transects were walked 2–4 times per week, weather permitting, between 1 July and 15 August, except for 2009, when the monitoring terminated earlier, and the extraordinarily warm year 2018, with the first random records from June 7. Monitoring started earlier (15 June) during 2019 and 2020 to cover the possibility of an earlier emergence of adults. However, we did not record any adults before the beginning of July during these 2 years with extended monitoring period. The walks were restricted to air temperature > 15 °C, clear or half-overcast sky, and wind < 11 km/h. Permissions for the field research were obtained from the administration of the Jeseníky Protected Landscape Area (permit no. 442/JS/10) and the administration of the Krkonoše National Park (permit no. KRNAP 03599/2013).

## Abundance and phenology indices

We described the relative annual abundance using the population abundance index (*PAI*) (*Kleckova, Vrba & Konvicka, 2015*; *Konvicka et al., 2016*), a measure derived from the shapes of the population recruitment curves, defined by daily relative abundances related to day-of-year (*Rothery & Roy, 2001*) (Table S1). We fitted the curves (Figs. S1–S11) *via* the generalised additive models (GAMs) using mgcv (*Wood, 2011*) in R 3.4.4 (*R Development Core Team, 2018*), with cubic splines (k = 4) and quasi-Poisson errors. The number of daily records were standardised for unit transect length. To facilitate the convergence of the fitted curve to zero prior to and after the adult flight period, zero values

were added 1 week before the first monitoring record and 2 weeks after the last adults observation on the transect. The areas under the GAM curves were computed using the rectangle method, which approximates a definite integral. Specifically, the date interval between the added zeroes for each species and transect was divided into subintervals of length 0.01 days ($x$), predicted abundance value ($y$) was generated for each subinterval by the relevant GAM, and the areas of the resulting rectangles ($x * y$) were summed across the entire date interval to obtain the approximate area under the curve, *i.e.*, the population abundance index (*PAI*).

For phenology, we used the GAM-fitted population recruitment curves (Figs. S1–S11) and the function predict (library mgcv) to infer the days when 20% and 80% of individuals were in flight (an example script is available in Figshare at https://doi.org/10.6084/m9. figshare.14642472). The 20% day is herein the flight period *onset*, and 20–80% day interval length is the flight period *duration* (Table S1). This diminishes the uncertainty which would arise from using the dates of the first and last observed individual to describe the flight period duration.

## Climate variables

Monthly weather data were provided by the Czech Hydrometeorological Institute (https://www.chmi.cz/historicka-data/pocasi/mesicni-data). For J, 1995–1997 records were from the Mt Praděd meteorological station (1,491 m, N 50.0831° E 17.2310°) adjoining the transect J3, and 2009–2020 records from Mt Šerák station (1,328 m, N 50.1874° E 17.1082°), located 15 km NW, but climatically equivalent to the study sites. For K (2010–2020), we used records from the Luční Bouda station (1,410 m, N 50.7344° E 15.6973°) for the summit transects K3 and K4, and from the Pec pod Sněžkou station (820 m, N 50.6918° E 15.7287°) for the lower-altitude transects K1 and K2.

Adult abundance and phenology may be affected by weather experienced by the immatures (*Turlure et al., 2010*), including extremes such as ground frosts. For the pre-hibernation larvae (September–October), we considered total autumn precipitation ($P_{tot}Aut$), as well as monthly average ($T_{avg}Aut$), maximum ($T_{max}Aut$), minimum ($T_{min}Aut$), and ground minimum ($T_{ground}Aut$) temperatures, averaged across the 2 months (Table S1). For the overwintering (November–April), when the mountains are usually snow-covered, we used the variables as above (indexed $P_{tot}Win$, $T_{avg}Win$, $T_{max}Win$, $T_{min}Win$, $T_{ground}Win$) and the snow cover duration (in days, $Snow_{days}$). For the spring larval period (May–June), we used the same predictors as for autumn (indexed $P_{tot}Spr$, $T_{avg}Spr$, $T_{max}Spr$, $T_{min}Spr$, $T_{ground}Spr$). The same applied for summer (July–August), when the weather directly affects the adults ($P_{tot}Sum$, $T_{avg}Sum$, $T_{max}Sum$, $T_{min}Sum$, $T_{ground}Sum$).

Whereas most of the climate variables were available for all years in J (1995–1997 + 2009–2020, herein J$_{all}$) the monthly number of snow days ($Snow_{days}$) and ground minimum temperature ($T_{ground}$) were available only for the J recent period (2009–2020), J$_{rec}$. The ground minimum temperature, measured at 5 cm above the ground, provides a proxy for days with ground frost, possibly critical for larval survival (*Vrba, Konvicka & Nedved, 2012*). $T_{ground}$ and $Snow_{days}$ were not available for K.

We calculated the Pearson's correlation coefficients for all climate variables for $J_{all}$, $J_{rec}$, K lower altitudes (transects K1, K2), and K higher altitudes (K3, K4), and visualised the matrices using the package corrplot (*Dray, 2008*) (Figs. S12–S15). We also computed the Principal Component Analysis (PCA) in *vegan* for R (*Oksanen et al., 2019*) (Fig. S16). These analyses revealed multiple correlations among the climate variables, but without a clear general pattern.

## Statistical analyses

We conducted a separate analysis for each mountain range and for the two periods in J: $J_{all}$ and $J_{rec}$. We ran the analyses separately for the two species in $J_{rec}$. We first tested whether there was a temporal trend in the abundance and phenology of the butterflies, and in the climate variables. We used generalised linear models (GLM) in R to test the linear effect of year on the response variables, with transect identity as a fixed effect. We also tested for possible biennial abundance fluctuations in population abundance index of *E. sudetica* and *E. epiphron* using a GLM with odd *vs* even year as a predictor, including the transect identity and linear effect of the year as covariates. We similarly tested whether *PAI* depended on *Onset* and *Duration*, and whether the two phenological variables were correlated.

For the effect of climate predictors on *PAI*, *Onset* and *Duration*, we tested the effects of conditions experienced by autumn, overwintering, and spring larvae, and the effect of conditions experienced by adults during their flight period on *PAI* and *Duration*. Separate analyses were performed for four datasets: *E. sudetica*, *E. epiphron* K, *E. epiphron* $J_{rec}$, and *E. epiphron* $J_{all}$ to account for differences in the availability of climate variables between J and K, and between $J_{rec}$ and $J_{all}$. Analysing *E. epiphron* data separately for J and K allowed detection of possible inter-population differences; separate analyses of data from $J_{rec}$ and $J_{all}$ also facilitated the comparison of trends in *E. epiphron* and in *E. sudetica*, because the latter was monitored only during the recent period (2009–2020).

We used an information theoretic approach, which is more appropriate for comparing large numbers of models with different predictors than a frequentist hypothesis testing (*Symonds & Moussalli, 2011*). Understanding that climate variables are always causally interdependent and numerically intercorrelated, and given the rather low degrees of freedom (3/4 transects for J/K × 11/12/15 monitoring years for K/$J_{rec}$/$J_{all}$), we did not fit models with multiple predictors and their interactions, as this would risk overfitting and unreliable estimates (*Harrell, 2015*). Instead, we evaluated individual predictors separately.

We used GLM with population abundance index (PAI, log-transformed) per year and transect as the response variable and transect as a fixed factor. The predictors were standardised to zero mean and unit variance, which enabled relative comparisons of their effects. The residuals were checked against the predicted values using simulateResiduals in DHARMa for R (*Hartig, 2016*).

We first constructed, for each of the four data sets, a null model not containing predictors: $log(PAI) \sim 1 + Transect$. Its *AIC* was used to evaluate subsequently fitted models according to $\Delta AIC$ between the null and a fitted model(s). We considered $\Delta AIC < -6$ a strong and $< -2$ a moderate indication of an effect of the respective predictor

**Table 1 The dependence of the population abundance index (*PAI*) on the flight period *Onset* and *Duration*.**

| Model | E. sudetica Jeseník Mts 2009–2020 (J) | | | | E. epiphron Jeseník Mts 1995–1999 + 2009–2020 (Jall) | | | | E. epiphron Jeseník Mts 2009–2020 (Jrec) | | | | E. epiphron Krkonoše Mts 2010–2019 (K) | | | |
|---|---|---|---|---|---|---|---|---|---|---|---|---|---|---|---|---|
| | ΔAIC | Slope | SE | df | ΔAIC | Slope | SE | df | ΔAIC | Slope | SE | df | ΔAIC | Slope | SE | df |
| Log(PAI) ~ Transect | 10.6 | – | – | 33 | 15.2 | – | – | 38 | 13.1 | – | – | 33 | **0.0** | – | – | **36** |
| Log(PAI) ~ Transect + log(Onset) | 8.3 | −4.79 | 2.38 | 32 | 13.7 | −4.23 | 2.34 | 37 | 11.9 | −4.31 | 2.49 | 32 | 2.0 | 0.39 | 2.52 | 35 |
| Log(PAI) ~ Transect + log(Duration) | **0.0** | **0.79** | **0.22** | **32** | **0.0** | **0.74** | **0.17** | **37** | **0.0** | **0.88** | **0.22** | **32** | 0.6 | −0.33 | 0.29 | 35 |
| Log(PAI) ~ Transect + log(Onset) + log(Duration) | 1.9 | – | – | 31 | 1.6 | – | – | 36 | 1.6 | – | – | 31 | 2.0 | – | – | 34 |

Note:
ΔAIC is compared to the best-fitting model (bold). All models also contain the effect of transect identity.

(*Richards, Whittingham & Stephens, 2011*). Because population abundances may reflect dependence on previous years, we tested for the presence of a temporal autocorrelation using the corAR1 function (autocorrelation structure of order 1). For all data sets, the null models without autocorrelation had smaller *AIC*s than those with autocorrelation, and we did not consider autocorrelation further.

We used an analogous procedure for flight period *Onset* and *Duration*. We did not consider autocorrelation with the previous year, as there is no reason to expect dependency of flight timing on the previous year abundance.

## RESULTS

In J/K, we walked 544/646 transect walks (mean ± SD: 32 ± 14.8/65 ± 14.4), obtaining 3,827 records of *E. sudetica*, 12,137 records of *E. epiphron* J, and 14,010 records of *E. epiphron* K (Supporting Information, Table S1). Raw data are available in Figshare at https://doi.org/10.6084/m9.figshare.14642472.

### Correlations among variables

We found a strong negative correlation between the flight period *Onset* and *Duration* in *E. sudetica* ($r = -0.70$, 95% confidence interval = $[-0.84, -0.48]$), *E. epiphron*, J (Jall: $r = -0.73$, $[-0.84, -0.53]$; Jrec: $r = -0.71$, $[-0.84, -0.50]$) and K ($r = -0.64$, $[-0.79, -0.41]$). Thus, the earlier onset resulted in longer duration of the flight period. In *E. sudetica* and *E. epiphron* Jeseniky Mts., the relative abundance (*PAI*) significantly increased with *Duration* but was not affected by *Onset* (Table 1, Fig. S17). Some of the climate variables were correlated, in particular different temperature variables within individual parts of the year (*e.g.*, $T_{avg}Win$ and $T_{min}Win$) (Supporting information, Figs. S12–S16).

### Temporal trends

We detected several temporal trends in climate variables over the entire study period in Jall (*i.e.*, including the 1990s data), but most of them were not apparent over the shorter recent period (Jrec and K). Specifically, most measures of temperature increased in Jall, winter precipitation increased in Jrec, and summer precipitation decreased in K. The trends in

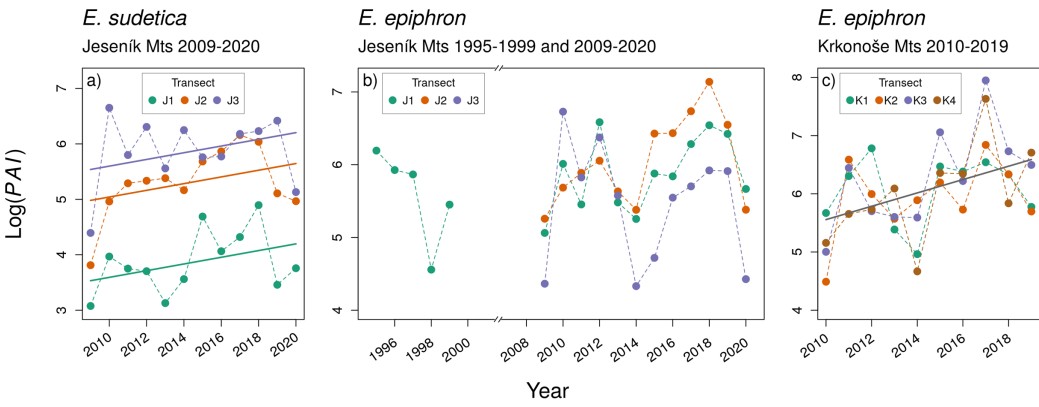

**Figure 2** The temporal changes in the abundance index (PAI) of (A) *Erebia sudetica* and (B–C) *E. epiphron*. Colours denote individual transects.

**Table 2** Temporal trends in the population abundance index (*PAI*) and flight period *Onset* and *Duration*.

| Model | *E. sudetica* J 2009–2020 | | | | *E. epiphron* J_all 1995–1999+ 2009–2020 | | | | *E. epiphron* J_rec 2009–2020 | | | | *E. epiphron* K 2010–2019 | | | |
|---|---|---|---|---|---|---|---|---|---|---|---|---|---|---|---|---|
| | *ΔAIC* | Slope | SE | *df* | *ΔAIC* | Slope | SE | *df* | *ΔAIC* | Slope | SE | *df* | *ΔAIC* | Slope | SE | *df* |
| *Population abundance index (PAI)* | | | | | | | | | | | | | | | | |
| Log(PAI) ~ Transect | 3.0 | – | – | 33 | **0.0** | – | – | **38** | **0.0** | – | – | **33** | 8.1 | – | – | 36 |
| Log(PAI) ~ Transect + Year | **0.0** | **0.060** | **0.028** | **32** | 0.3 | 0.020 | 0.017 | 37 | 0.2 | 0.039 | 0.031 | 32 | **0.0** | **0.115** | **0.036** | **35** |
| *Flight period onset* | | | | | | | | | | | | | | | | |
| Log(Onset) ~ Transect | **0.0** | – | – | **33** | **0.0** | – | – | **38** | **0.0** | – | – | **33** | 1.6 | – | – | 36 |
| Log(Onset) ~ Transect + Year | 0.7 | −0.002 | 0.002 | 32 | 1.0 | −0.001 | 0.001 | 37 | 1.1 | 0.002 | 0.002 | 32 | **0.0** | **−0.005** | **0.003** | **35** |
| *Flight period duration* | | | | | | | | | | | | | | | | |
| Log(Duration) ~ Transect | 0.3 | – | – | 33 | 11.0 | – | – | 38 | **0.0** | – | – | **33** | **0.0** | – | – | **36** |
| Log(Duration) ~ Transect + Year | **0.0** | **0.029** | **0.020** | **32** | **0.0** | **0.042** | **0.011** | **37** | 1.0 | 0.020 | 0.021 | 32 | 0.2 | 0.029 | 0.023 | 35 |

**Note:**
*ΔAIC* is compared to the best-fitting model (bold); separately for *PAI*, *Onset*, and *Duration*.

these variables were similar in both mountains, but often weak in terms of statistical significance (Table S2).

Despite large inter-annual variation, adult abundance significantly increased in *E. sudetica* J and *E. epiphron* K, but not in *E. epiphron* J (Table 2, Fig. 2). There was no temporal trend in *Onset* (Fig. S18) and in most cases in *Duration* (Fig. S19, Table 2) with the exception of *E. epiphron* J_all, for which the flight period *Duration* increased since the 1990s (Fig. S19). *E. epiphron* K displayed an evidence of biennial fluctuations, with population abundance index ≈ 1.8 times higher in odd years than in even years (GLM, the model including the odd/even year had *ΔAIC* = −4.7 compared to the null model), but no such pattern applied for either of the species in J.

## Climate variables and abundance

In *E. sudetica*, warmer autumns ($T_{min}Aut$, $T_{ground}Aut$), higher winter precipitation ($P_{tot}Win$) and higher spring temperatures ($T_{min}Spr$, $T_{max}Spr$, $T_{avg}Spr$) increased *PAI* in the

**Table 3 The effect of the climate predictors, each tested separately, on the population abundance index (PAI).**

| Model | E. sudetica J 2009–2020 (Null df = 33) | | | E. epiphron J$_{all}$ 1995–1997 + 2009–2020 (Null df = 38) | | | E. epiphron J$_{rec}$ 2009–2020 (Null df = 33) | | | E. epiphron K 2010–2019 (Null df = 36) | | |
|---|---|---|---|---|---|---|---|---|---|---|---|---|
| | ΔAIC | Slope | SE | ΔAIC | Slope | SE | ΔAIC | Slope | SE | ΔAIC | Slope | SE |
| Null model | 0.0 | | | 0.0 | | | 0.0 | | | 0.0 | | |
| $P_{tot}Aut$ | 1.99 | 0.01 | 0.10 | 1.61 | −0.06 | 0.10 | 1.69 | −0.06 | 0.11 | −1.90 | 0.22 | 1.89 |
| $T_{avg}Aut$ | 1.59 | 0.06 | 0.10 | 1.59 | 0.06 | 0.25 | 1.22 | 0.09 | 0.11 | −1.37 | 0.34 | 0.19 |
| $T_{max}Aut$ | 0.91 | −0.10 | 0.10 | 1.67 | −0.06 | 0.10 | 1.64 | −0.06 | 0.11 | 1.86 | 0.05 | 0.13 |
| $T_{min}Aut$ | **−3.12** | **0.21** | **0.10** | **−5.70** | **0.26** | **0.09** | **−9.08** | **0.33** | **0.10** | **−6.23** | **0.55** | **0.20** |
| $T_{ground}Aut$ | **−5.77** | **0.26** | **0.09** | – | – | – | **−11.08** | **0.35** | **0.09** | – | – | – |
| $P_{tot}Win$ | **−6.99** | **0.28** | **0.09** | **−11.25** | **0.33** | **0.09** | **−10.14** | **0.34** | **0.09** | **−7.80** | **0.65** | **0.21** |
| $T_{avg}Win$ | 1.33 | −0.08 | 0.10 | **−4.61** | **−0.25** | **0.10** | **−6.01** | **−0.28** | **0.10** | 1.91 | −0.06 | 0.20 |
| $T_{max}Win$ | −1.42 | 0.18 | 0.10 | 1.89 | 0.03 | 0.10 | 1.62 | 0.07 | 0.11 | 1.74 | 0.10 | 0.20 |
| $T_{min}Win$ | 1.47 | −0.07 | 0.10 | −1.97 | −0.19 | 0.10 | **−3.13** | **−0.23** | **0.10** | 1.79 | 0.08 | 0.18 |
| $T_{ground}Win$ | 1.12 | −0.09 | 0.10 | – | – | – | **−4.01** | **−0.25** | **0.10** | – | – | – |
| $Snow_{days}$ | 1.75 | 0.05 | 0.10 | – | – | – | −1.47 | 0.19 | 0.11 | – | – | – |
| $P_{tot}Spr$ | **−5.74** | **−0.26** | **0.09** | −1.74 | −0.18 | 0.10 | −1.68 | −0.20 | 0.11 | −1.78 | −0.22 | 0.12 |
| $T_{avg}Spr$ | **−7.30** | **0.28** | **0.09** | **−14.01** | **0.36** | **0.09** | **−22.55** | **0.45** | **0.08** | **−4.01** | **0.67** | **0.28** |
| $T_{max}Spr$ | **−5.91** | **0.26** | **0.09** | **−3.52** | **0.22** | **0.10** | **−4.65** | **0.26** | **0.10** | 1.97 | 0.04 | 0.23 |
| $T_{min}Spr$ | **−3.58** | **0.22** | **0.10** | **−2.45** | **0.20** | **0.10** | **−3.96** | **0.25** | **0.10** | 1.58 | −0.14 | 0.23 |
| $T_{ground}Spr$ | 0.09 | 0.13 | 0.10 | – | – | – | 0.38 | 0.13 | 0.11 | – | – | – |
| $P_{tot}Sum$ | 0.93 | −0.10 | 0.10 | 2.00 | 0.00 | 0.10 | 2 | 0.00 | 0.11 | **−5.19** | **−0.29** | **0.11** |
| $T_{avg}Sum$ | −0.50 | 0.15 | 0.10 | 1.63 | 0.06 | 0.11 | 1.14 | 0.10 | 0.11 | 0.90 | 0.26 | 0.26 |
| $T_{max}Sum$ | 1.85 | −0.04 | 0.10 | 1.89 | 0.03 | 0.11 | 1.67 | 0.06 | 0.11 | −0.40 | 0.24 | 0.16 |
| $T_{min}Sum$ | **−5.21** | **−0.25** | **0.09** | −1.89 | −0.19 | 0.10 | **−2.7** | **−0.22** | **0.11** | 1.89 | 0.10 | 0.32 |
| $T_{ground}Sum$ | −0.93 | −0.16 | 0.10 | – | – | – | **−3.32** | **−0.24** | **0.10** | – | – | – |

**Note:**
The predictors were standardised to have zero mean and unit variance. The response variable (PAI) was log-transformed. ΔAIC = the difference of AIC of each listed model containing a single climate variable, along with the effect of transect identity, compared to the corresponding null model without the effect of the climate variable; i. e., log(PAI) ~ Transect. Hence, models with lower AIC compared to the null model have negative values of ΔAIC. ΔAIC < −6 was considered a strong (bold underlined) and < −2 a moderate (bold) indication of the respective predictor's effect. Empty cells (–): the predictor not available for the mountain range and time period.

following year. In contrast, increasing spring precipitation ($P_{tot}Spr$) and increasing minimum summer temperatures ($T_{min}Sum$) decreased population abundance index in the following year (Table 3).

For *E. epiphron* J, adult population abundance index increased with autumn temperatures ($T_{min}Aut$, $T_{ground}Aut$), winter precipitation ($P_{tot}Win$) and spring temperatures ($T_{avg}Spr$, $T_{max}Spr$, $T_{min}Spr$), but decreased with winter temperatures in ($T_{avg}Win$) (Table 3) and higher summer minimum ground temperatures ($T_{ground}Sum$). The negative effect of winter temperatures ($T_{avg}Win$, $T_{min}Win$, $T_{ground}Win$) was more distinct for J$_{rec}$ than for J$_{all}$ (Table 3). For J$_{rec}$, $T_{min}Win$ and $T_{min}Sum$ displayed moderate effects, not detected for J$_{all}$.

For *E. epiphron* K, the predictors increasing subsequent population abundance index consistently with *E. epiphron* J were higher autumn temperatures ($T_{min}Aut$), higher winter

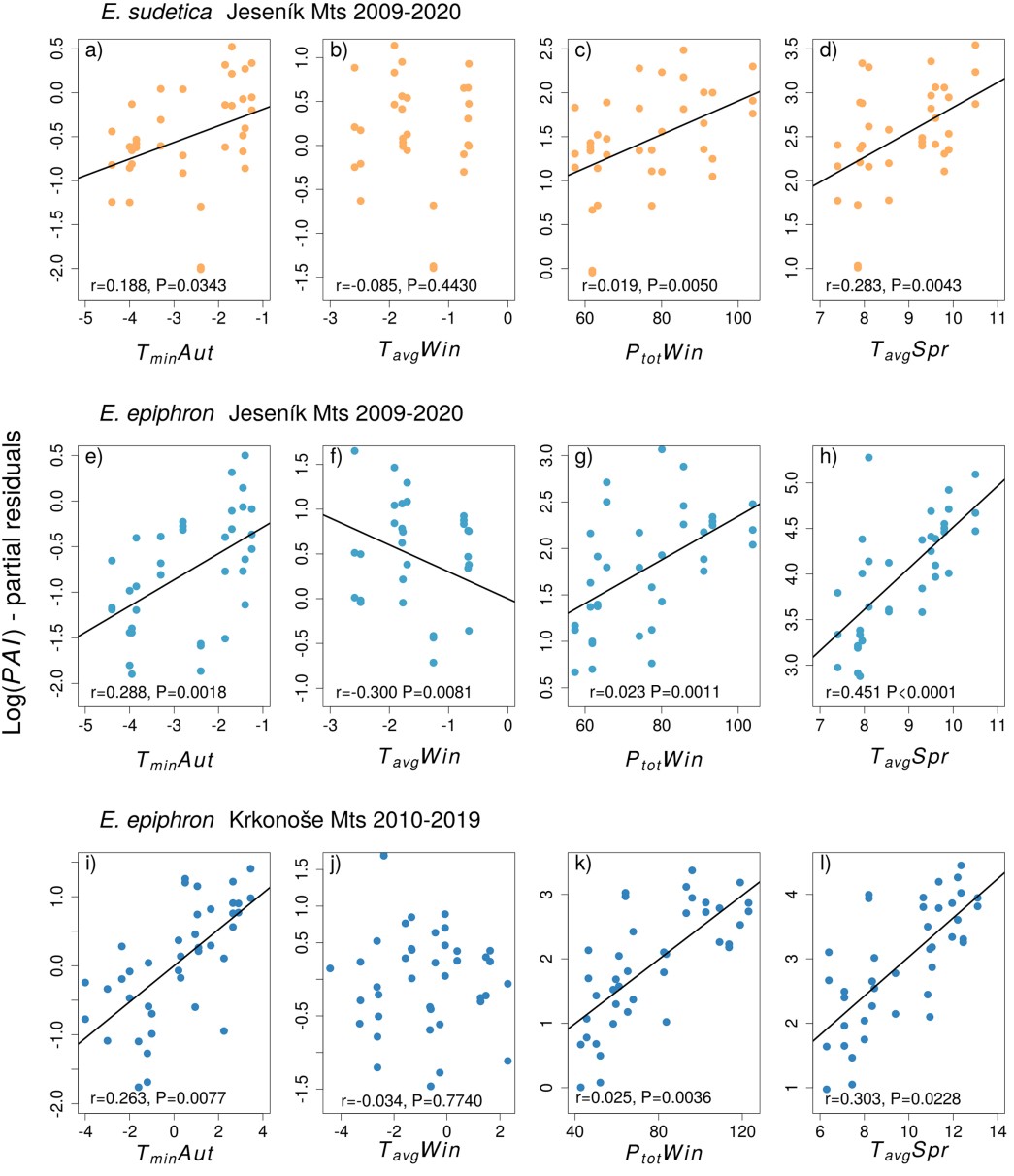

**Figure 3** **The effect of climate experienced by larvae during the autumn (Aut), winter (Win), and spring (Spr) on subsequent adult abundances of (A–D)** *Erebia sudetica* **and (E–L)** *E. epiphron* **(the PAI index, see Methods for details).** Climate variables (T = temperature, P = precipitation) are shown at their original scale (not standardised) to make the plots more informative. Only selected climate variables are displayed; parameter estimates for all variables are shown in Table 3. See the Methods section for detail description of the climate variables.

precipitation ($P_{tot}Win$), and higher spring temperatures ($T_{avg}Spr$) (Fig. 3). The negative effect of warm winters, prominent in J, was not detected in K. Another difference was higher precipitation in summer ($P_{tot}Sum$), reducing adult abundances in J ($\Delta AIC < -2$) but not in K. The major trends of warm springs increasing adult abundance, and winters with little precipitation decreasing it, were consistent for the two mountains (Table 3).

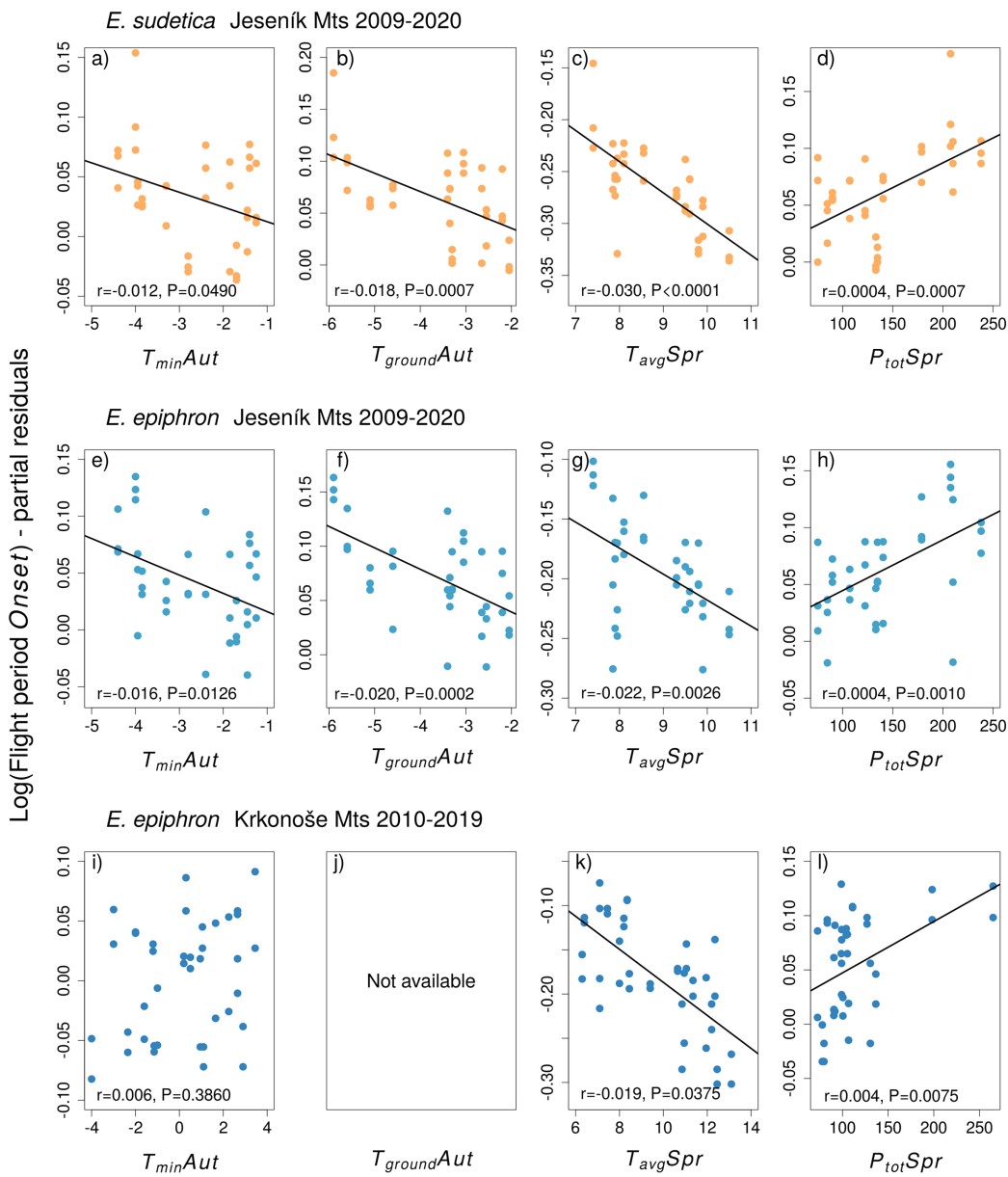

**Figure 4** **The effects of the climate variables (T = temperature, P = precipitation) experienced by larvae during the autumn (Aut) and spring (Spr) on the onset of the flight period of (A–D)** *Erebia sudetica* **and (E–L)** *E. epiphron.* Climate variables are shown at their original scale (not standardised) to make the plots more informative. Only selected climate variables are displayed; parameter estimates for all variables are shown in Table S3. See the Methods section for detailed description of the climate variables.

## Climate variables and phenology

Adult phenology of *E. sudetica* (Table S3 and S4, Figs. 4 and 5) was affected by climate in all larval phases. Higher autumn ground temperatures ($T_{ground}Aut$) advanced *Onset*, *i.e.*, accelerated larval development. Higher spring precipitation ($P_{tot}Spr$) postponed *Onset* and shortened *Duration*. Warm springs ($T_{avg}Spr$, $T_{max}Spr$, $T_{min}Spr$) advanced *Onset* and prolonged *Duration*. Summer weather had no effect on *Duration* (Table S3).

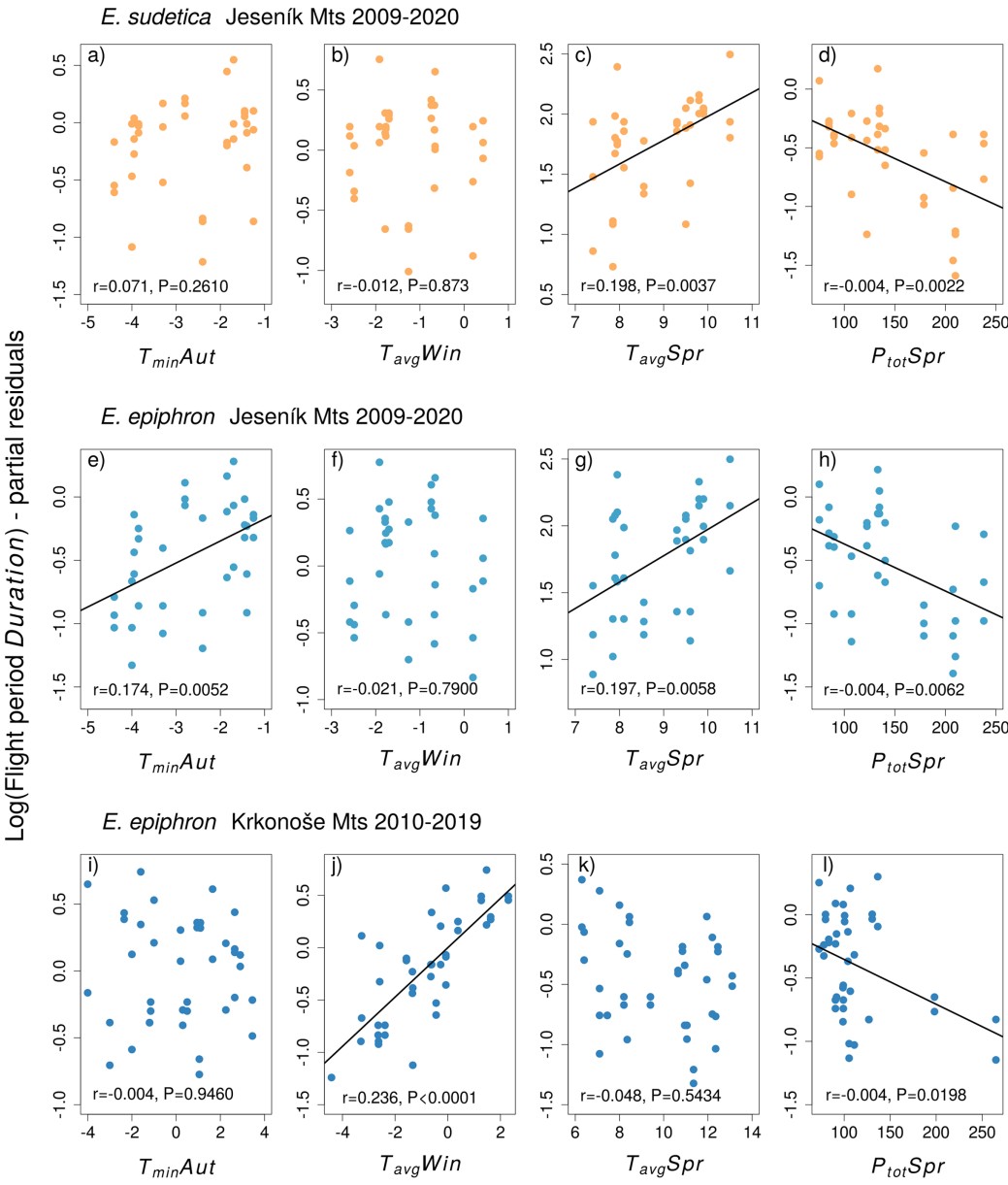

**Figure 5 The effects of the climate variables (T = temperature, P = precipitation) experienced by larvae during the autumn (Aut) and spring (Spr) on the duration of the flight period of (A–D) *Erebia sudetica* and (E–L) *E. epiphron*.** Climate variables are shown at their original scale (not standardised) to make the plots more informative. Only selected climate variables are displayed; parameter estimates for all variables are shown in Table S4. See the Methods section for detailed description of the climate variables.

In *E. epiphron* J, warmer autumns ($T_{min}Aut$, $T_{ground}Aut$) and springs ($T_{avg}Spr$) advanced *Onset* (Tables S3, Fig. 4) and prolonged *Duration* (Table S4). Higher spring precipitation ($P_{tot}Spr$) had opposite effects. In summer, flight period *Duration* increased with $T_{avg}Sum$.

In *E. epiphron* K, the factors affecting phenology partly differed from J, but there were commonalities (Table S3, Figs. 4, 5). As in J, rainy springs ($P_{tot}Spr$) delayed *Onset* and shortened *Duration*, whereas warm springs ($T_{avg}Spr$, $T_{min}Spr$) advanced *Onset*. Contrary

to J, warm autumns ($T_{max}Aut$) delayed and rainy autumns ($P_{tot}Aut$) advanced *Onset* (Table S3). In addition, warm winters ($T_{avg}Win$, $T_{min}Win$) and summers ($T_{max}Sum$) shortened *Duration*, which was not detected in J (Table S4).

## DISCUSSION

Fitting indices of the abundance and phenology of two subalpine butterflies obtained by a decade of detailed monitoring to climate data revealed patterns remarkably consistent between *E. epiphron*, a species of wind-exposed flat summit grasslands, and *E. sudetica*, a species associated with tall-herb vegetation of leeward sites near the timberline. For the former, the patterns were also mostly consistent between two mountain systems, isolated by more than 100 km. In both species, higher winter temperatures and lower precipitation (*i.e.*, less snow) experienced by overwintering larvae diminish subsequent adult abundance (cf. *Rytteri, 2021*), whereas warmer autumns, and warmer and drier springs, increase it (Table 3, Fig. 4). Warm autumns and springs also advance the adult flight onset, which is positively linked to flight duration (Table S3). These observations were made against a background of generally increasing temperatures (*Migala, Urban & Tomczyński, 2016*) in both mountain ranges, and against stable (*E. epiphron*, J) or increasing (*E. sudetica*, J, *E. epiphron*, K) population trends (Fig. 2).

The results have profound implications for the studied species and mountain insects in general. Warmer autumns and springs appear beneficial for subsequent adult abundances, presumably *via* increased larval survival. These effects are counteracted by low winter precipitation and high winter temperatures, implying low snow cover.

For hibernating insects, lack of snow worsens the insulation of the ground layer, exposing them to deep frosts (*Huang, 2016*). Interestingly, *Vrba, Konvicka & Nedved (2012)* reported that *E. epiphron* and *E. sudetica* had higher values of lower lethal temperatures than their lowland congeners. This suggests that they may be more sensitive to frosts, because they are adapted to predictable winter conditions in the mountains where the ground is insulated by snow for the entire winter, unlike in the lowlands where frosty periods without snow are common (*Vrba, Konvicka & Nedved, 2012*). Warm periods during winter may increase overwintering insect mortality due to fungal infections (*Steenberg & Ogaard, 2000*), diapause breakups (*Lindestad et al., 2020*), or metabolic failures (*Klockmann & Fischer, 2019*). Overwintering without snow in combination with advanced spring onset decreased overwintering survival in *Melitaea cinxia* (*Rytteri, 2021*). However, *E. epiphron* and *E. sudetica* were not affected by snow cover duration ($Snow_{days}$) based on our results (Table 3).

Warm autumns and springs had opposite effects than warm winters, increasing adult abundances, advancing adult flight onsets and extending their durations. Warm conditions during larval feeding periods may accelerate the development (*Roy et al., 2001*) and shorten the exposure to natural enemies (*Culler, Ayres & Virginia, 2015*). Increased pre-hibernation food intake may improve the larval energy budget, allowing synthesis of more cryoprotective compounds (*Vrba et al., 2017a*). An identical logic likely applies to post-hibernation larvae, in which the warmer and drier weather accelerates development, advancing the flight onsets (*Gutiérrez & Wilson, 2020*; *Stewart et al., 2020*).

The beneficial effects of warmer temperatures apply only up to species-specific limits, set by heat tolerance (*Terblanche et al., 2017*), diapause/moulting triggers (*Gonzalez-Tokman et al., 2020*), or thermal effects mediated *via* host plants (*Barrio, Bueno & Hik, 2016*). Caterpillar feeding may be impaired by temperatures above their thermal limits (*Klockmann, Wallmeyer & Fischer, 2018*), and feeding efficiency may decline with increased respiration rates (*Bauerfeind & Fischer, 2013*; *Kukal & Dawson, 1989*). As we found no negative effects of warm autumns or springs, we presume that the warmer conditions of the last decade had not yet exceeded the studied species' thermal optima.

During adult flight, high minimum temperatures decreased both species' abundances. This puzzling effect is likely linked to the summer mountain climate with cold nights and mornings during high-pressure periods (warm days), and warm nights and mornings during foggy low-pressure periods (cold days). Foggy weather leaves insects defenceless to warm-blooded predators (birds, small mammals) and may cause starvation of adults. Weather effects on lepidopteran adult longevity and population dynamics remain little explored (*cf. Bubova et al., 2016*), and deserve attention in mountain environments (*Junker et al., 2010*).

The mechanisms extending flight duration following warmer larval conditions (Table S4) appear less straightforward. Univoltine butterflies of Sierra de Guadarrama, Spain, advanced flight onset after warm springs, but this was not always connected with increased abundances (*Stewart et al., 2020*). *Zografou et al. (2020)* observed delayed flight onsets and shorter durations in higher elevations of two Greek mountain systems. In the *Erebia* butterflies studied here, warmer conditions during larval feeding periods may support larval growth, extending adult longevity. Alternatively, warmer conditions may desynchronise larval development, causing gradual adult emergence in low densities rather than abrupt emergence in high densities (*cf. Ehl et al., 2017*). Also recall that *Erebia* spp. can be flexible regarding the larval development duration (*Wipking & Mengelkoch, 1994*). The development of *E. epiphron* lasted a single season in lowland outdoor conditions (*Kuras et al., 2001*), but it takes two seasons in the Alps (*Sonderegger, 2005*). We did not detect indications of biennial development in J, but detected them in K. The latter mountains are only ≈100 m higher, but are situated more westerly, exposed directly to winds blowing from the North Sea over the German Plains. Still, the *E. epiphron* biannual fluctuations in K were less prominent than in a related species *E. euryale* with prevailing biennial development (*Kleckova, Vrba & Konvicka, 2015*). Possibly, the development length varies among individuals, lasting 1 or 2 years according to the local conditions. If so, warm weather during larval periods may allow for pupation in some larvae that would otherwise develop for an additional season. These butterflies would emerge relatively late in the flight period, optically prolonging its duration. The positive relationship between flight duration and population abundance index call for the latter possibility.

The positive effects of increasingly warm autumns and springs likely counteract the negative effects of snowless winters on the *Erebia* butterflies' population dynamics. Remarkably similar patterns apply to other alpine insects. *Roland & Matter (2016)* found,

for the butterfly *Parnassius smintheus* Doubleday, 1847 in the Canadian Rocky Mts, negative effects of November temperature extremes, and positive effects of warm springs. November (*i.e.*, early winter) snow cover was the best predictor of the following seasons' abundances (*Roland, Filazzola & Matter, 2021*). Climate effects differing across species or phenotypes were demonstrated by *Buckley & Kingsolver (2012)*, who modelled the interaction between the time available for flight and egg viability in two alpine *Colias* spp. in the Rocky Mts, USA. In cooler conditions, the two species differed in their ability to respond by extension of the flight period, which should balance the decreased egg viability caused by weather extremes. Beyond insects, *Chirichella et al. (2020)* found that better forage availability during warm winters increases the reproductive output of the alpine chamois (*Rupicapra rupicapra*), but this effect was outweighed by the negative effects of decreased forage quality during warm summers. Outside of high mountains, *Long et al. (2017)* detected warm winters as detrimental and hot summers as beneficial for the majority of British butterflies, whereas cold winters and cold summers had the opposite effects. All these studies warn against simplistic assumptions of climate effects on alpine fauna based on single measures, such as mean annual temperatures.

Warmer and drier autumns and springs allow building up population numbers, likely buffering the populations against the detrimental effects of snowless winters. The increasing yearly population numbers of *E. sudetica* and *E. epiphron* suggest that the positive effects of the warmer vegetation season currently prevail. However, we observed a notable drop in their numbers in 2020, a year with a low amount of snow in the winter followed by a rainy summer (Fig. 2). Population numbers are thus reflecting ongoing climatic change (warmer vegetation season) and stochastic fluctuations of local precipitation (*Migala, Urban & Tomczyński, 2016*). The effect of these factors is complex and may be driven by interactions between precipitation and temperature (*Rytteri, 2021*). Longer-term monitoring data will be needed to provide more detailed insight.

As a rule, climate variation is buffered by diverse microclimates in mountain environments (*Nieto-Sanchez, Gutierrez & Wilson, 2015*; *Roland, Filazzola & Matter, 2021*; *Turlure et al., 2010*; *Wilson et al., 2015*) and the more abundant a population, the more individuals will likely locate microclimatically suitable sites. This provides grounds for moderate optimism regarding the future of the studied populations and other insects of temperate zone middle-high mountains. In addition, in both the Jeseník and Krkonoše mountains, the timberline elevation had fluctuated during the Holocene, rising to higher elevations than at present, *e.g.*, during the Atlantic period, and descending due to booming pastoralism in the early modern era (*Treml et al., 2016*). The studied butterflies persisted through these fluctuations (*Schmitt, Cizek & Konvicka, 2005*), arguably owing to non-trivial population-level responses to fluctuating climate and habitat availability. However, while microclimate variability may facilitate the survival of these mountain populations, the availability of suitable habitats will strongly depend on the management of mountain grasslands, such as appropriate levels of grazing and suppression of expanding *Pinus mugo* brushwood (*cf. Bila et al., 2016*; *Zeidler et al., 2021*).

## CONCLUSIONS

We used data from long-term monitoring of mountain butterflies to understand the effect of climate change on mountain biological systems. The adult population abundances are driven by conditions experienced during their development. Warmer autumns and springs as well as high-precipitation winters experienced by larvae were reflected by higher adult population numbers of adult butterflies in subsequent summers. The adult population abundances were stable or even increasing in our study system. Thus, the positive and negative effects of climatic conditions experienced during development are in balance. We highlight that the observed positive population trends could be misleading as several years with adverse combinations of climatic conditions can lead to the collapse of local populations surviving on relatively small areas in low-elevation mountains. The long-term survival of mountain invertebrates can be supported by careful management, particularly by blocking of the encroachment of subalpine and alpine grasslands by trees and shrubs.

## ACKNOWLEDGEMENTS

We are grateful to dozens of students of the University of South Bohemia and volunteers for participation in the monitoring, the Jeseník Protected Landscape Area and Krkonoše National Park for logistic support, M. Sweney for linguistic corrections and the Czech Hydrometeorological Institute for climate data.

### Funding

This study was supported by the Czech Science Foundation (GA14-33733S, 18-23794Y), the Technology Agency of the Czech Republic (SS01010526), and the European Research Council (StG BABE 805189). The funders had no role in study design, data collection and analysis, decision to publish, or preparation of the manuscript.

### Grant Disclosures

The following grant information was disclosed by the authors:
Czech Science Foundation: GA14-33733S, 18-23794Y.
Technology Agency of the Czech Republic: SS01010526.
European Research Council: StG BABE 805189.

### Competing Interests

The authors declare that they have no competing interests.

### Author Contributions

- Martin Konvicka conceived and designed the experiments, performed the experiments, analyzed the data, prepared figures and/or tables, authored or reviewed drafts of the paper, and approved the final draft.

- Tomas Kuras performed the experiments, authored or reviewed drafts of the paper, and approved the final draft.
- Jana Liparova performed the experiments, authored or reviewed drafts of the paper, and approved the final draft.
- Vit Slezak performed the experiments, authored or reviewed drafts of the paper, and approved the final draft.
- Dita Horázná performed the experiments, authored or reviewed drafts of the paper, and approved the final draft.
- Jan Klečka performed the experiments, analyzed the data, prepared figures and/or tables, authored or reviewed drafts of the paper, and approved the final draft.
- Irena Kleckova conceived and designed the experiments, performed the experiments, analyzed the data, prepared figures and/or tables, authored or reviewed drafts of the paper, and approved the final draft.

## Field Study Permissions

The following information was supplied relating to field study approvals (*i.e.*, approving body and any reference numbers):

Permissions for the field research were obtained from the administration of the Jeseníky Protected Landscape Area (permit no. 442/JS/10) and the administration of the Krkonoše National Park (permit no. KRNAP 03599/2013).

## Data Availability

Raw data are available in Figshare:

Konvicka, Martin; Kuras, Tomáš; Liparova, Jana; Slezak, Vit; Horazna, Dita; Klečka, Jan; et al. (2021): Low winter precipitation, but not warm autumns and springs, threatens mountain butterflies in middle-high mountains. figshare. Dataset. https://doi.org/10.6084/m9.figshare.14642472.v2.

## Supplemental Information

Supplemental information for this article can be found online at http://dx.doi.org/10.7717/peerj.12021#supplemental-information.

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
