# Peer review of "Low winter precipitation, but not warm autumns and springs, threatens mountain butterflies in middle-high mountains"

_PeerJ, doi:10.7717/peerj.12021_

## Round 0.1 · original submission · Minor Revisions

Dear Dr. Kleckova and collaborators,

I am pleased to say that after independent reviews from three different reviewers, I believe your study is practically accepted for publication! Congratulations on this!!! Nonetheless, there are a few improvements that are needed before the formal acceptance of your manuscript.

Please prepare a revised version of your manuscript considering the minor changes required by the reviewers to deliver a new version of your study in a two-month period (August 16th, 2021, or at your earliest convenience). Please do not also forget to prepare a rebuttal letter informing your reviewers of all the performed changes. Finally, if you and your co-authors do not agree with any of the issues raised by the reviewers and do not want to perform the necessary changes, please do not forget to explain such a decision to the reviewers, providing your reasons for not doing the required changes.

Best regards,
Daniel Silva, Ph.D.

·

Basic reporting

No comment

Experimental design

No comment

Validity of the findings

No comment

Additional comments

General comments
The manuscript entitled “Low winter precipitation, but not warm autumns and springs, threaten mountain butterflies in middle-high mountains” explores a series of climatic factors that could shaped population trends of mountainous butterflies. It also connects changes in species abundance and their phenology through an interesting approach of seasonal variation on regional climatic conditions, assuming an effect on early life stages as well.
Overall, this is a well-written manuscript that I much enjoyed reading. Experimentally the study is well-designed and very well-performed, while the analyses are relevant and sampling size robust. The authors are aware of up-to-date literature and have a deep ecological understanding and scientific knowledge with regards to the paper scope.
This study is an outstanding contribution to the existing literature on phenological patterns and population trajectories because first, it provides data on cold adapted organisms that are less studied compared to their counterparts in lowlands and second, it gives access to long term data that are largely lacking from Central Europe.

I provide below some general thoughts and minor suggestions, hoping to be of use of the improvement of the paper.

Minor comments
84-85: Isn’t it the case for lowland species as well?
113-114: This part is somehow misleading. After reading it, I was expecting to see a combined field and lab work where all developmental stages of the species would have been monitored but it’s clearly not the case here. I’d suggest rewriting this part and try to include a term like “hypothetically all phases of larval development” or something similar.
164: Please correct the number of years. It is not 3 but 5.
168: Does this mean that you were in the field from month June, but you had no observations expect for the warm year 2018? If so, then I think you should need to clarify that as it increases the robustness of your sampling effort.
187: I think it might be easier to write population abundance index here. In general, I would try to minimize the use of acronyms as one can easily get lost especially when reading the methods and results.
188-193: It would be really useful to provide the script for onset and duration as well as the script for abundance index. I know that I would like to try your 20% (onset) and 20-80% (duration) approach in my next attempt to calculate phenological indices.
206: Please add the word “autumn” before total precipitation so that readers can understand what Aut stands for. Otherwise, you might want to replace “Aut” by the first letters of the two months such as SO (September, October). Please consider to add this explanation to the corresponding Table as well.
In addition, please replace “and” by “as well as”
207: Please delete “the”.
231: Please delete Rv.3.4.4 as it is mentioned earlier. I would suggest to report the version of R you used for your analysis either in the end of statistical methods or at the beginning of methods section. For example, “All statistical analysis were conducted in R v….”.
257: Please explain the four data sets, it is not clear here.
281-282: For all cases but E. epiphron in K mountain, right? Please correct.
288-289: Please consider including the whole time series as it might be more informative.
296: Please correct Table, it should be Table S4.
307: Is it again of the next year? Please clarify.
316-317: It might help if Table S3 was added here.
318-320: Maybe I am missing something here, but what I’m getting from Table S3 is certainly an increasing abundance during warm springs but not in summers (strong predictor effect has negative slope). With regards to winter precipitation, patterns seem to be increasing (positive slopes)
332: Please add “shortened” after the last word (and) of the sentence
350-351: It is not clear from Fig.3 that populations trends are stable for E.epiphron J. What about the decreasing trend in panel (f)?

Reviewer 2 ·

Basic reporting

No comment.

Experimental design

No comment.

Validity of the findings

No comment.

Additional comments

It is an important study, with interesting results to understand the effect of climate change on mountain biological systems. There is a considerable amount of data, the result of many years of monitoring. The analyzes are as expected for this type of study. The presentation of the figures and the results is done in a satisfactory way as well.

Some general comments:
I suggest adding a general conclusion, with the main points of the work.

As the authors use a large number of variables, it would be interesting to make the results clearer for readers, perhaps using a simple table (especially when presenting the relation between abundance and climatic variables), so the reader would be able to follow the discussion more easily.

There are some parts of the introduction that would fit in the methods, and of the discussion that is results. To avoid repeating the same information in the text, I would carefully review these two sessions.

Specific comments:
Line 54: “Middle-high”
Line 69: why not use “micro-habitat”?
Line 72: the same, why not use “sub-alpine”?
Line 106 to 110: would be better in the “methods” section.
Line 114: replace the phrase “We ask, which…” with “We tested which climate…”.
Line 133: “Permissions for the field…” would fit better in acknowledgment.
Line 136: Erebia sudetica does not occur then in J? Make this information clearer.
Line 163: “Together with students…” also would fit better in acknowledgment.
Line 168: The area is monitored in other months too then? It’s important to mention this, to show how the authors discovered that adults emerge before...
Line 229: And also separately for each species, correct?
Line 271: The information on the hours/sampling would also be relevant.
Line 320: Perhaps consider putting the tables S3 and S4 in the scope of the article, in some simplified way. It helps follow the text.
Line 345 to 348: This part is a bit of repetition of the results.
Line 358: higher or lower?
Line 366: Is this sentence correct “…were not affected by snow cover duration based on our results”? Because there was a snow cover effect, right?

·

Basic reporting

This is a well written manuscript of high relevance in the fields of ecology and conservation biology. All in all, it is well performed and I have only some relatively minor comments given at the general comments. Figures and tables are OK and the results allow the drawn conclusions. The discussion in general is interesting and relevant. The referring to the existing literature is well done and up-to-date.

Experimental design

Experimental disign is sound and well suitable to address the resarch question. The latter is well defined and relevant. The ms is performed at generally high standard.

Validity of the findings

The findings are solid and novel. They fulfil all necessary standards and thus fully merit publication in PeerJ.

Additional comments

L63: add 2with many of its species" after clade
L71: simplified --> lower
L85: Rephrase "advanced adult period onset". Maybe simply add flight after adult or exchange these two words.
L104: Add population after E. epiphron
L110: Rephrase, sounds odd.
Introduction in general: Maybe work out at the end the research question a bit more clearly.
L136: Not only Monts Du Cantal, but at several places (e.g. Puy Marie etc.)
L137: In Fagaras Mts, I also think in Rodna Mts, not sure abaout Retezat
L144: List incomplete; England must be northern England, missing Cantabrian Mts, Massif Central, Vosges, Central Apennines; maybe also mention the strong differentiation at nuclear (Schmitt, T., Hewitt, G.M. & Müller, P. (2006) Disjunct distributions during glacial and interglacial periods in mountain butterflies: Erebia epiphron as an example. Journal of Evolutionary Biology 19: 108-113.) and mt level (Minter, M., Dasmahapatra, K.K., Thomas, C.D., Morecroft, M.D., Tonhasca, A., Schmitt, T., Siozios, S. & Hill, J.K (2020) Past, current, and potential future distributions of unique genetic diversity in a cold‐adapted mountain butterfly. Ecology and Evolution 10: 11155–11168.).
L150: Normally end of June
L259: delta normally needs no "-" as it refers to a difference which is not negative
L344: over --> more than
L356: insulation --> isolation
L358: "higher lower lethal temperatures" sounds really odd; rephrase
L360: insulated --> covered
L368: periods --> flight
L400: indices --> indications
L435: precipitations --> precipitation
Figure 3, 4 and 5: Please add p and r in all cases in the figures.
Table 2: Why best model with negative delta AIC and not 0 with all others being positive?
In addition, I suggest to give the GLM results in one or two additional tables in the main ms.

---

## Round 0.2 · accepted · Accept

Dear Kleckova et al.,

After another review round, I am pleased to inform you that your manuscript was accepted for publication in PeerJ! Congratulations!

Sincerely,
Daniel Silva, Ph.D.

Reviewer 2 ·

Basic reporting

No comment

Experimental design

No comment

Validity of the findings

No comment

Additional comments

The authors accepted most of the suggestions, properly justifying those they did not. Thus, I consider the paper ready and suitable for publication.